# XIAP promotes the expansion and limits the contraction of CD8 T cell response through cell extrinsic and intrinsic mechanisms respectively

**Parva Thakker[1], Ardeshir Ariana[1], Stephanie Hajjar[1], David Cai[1], Dikchha Rijal[1], Subash Sad** [1,2]*

**1** Department of Biochemistry, Microbiology and Immunology, Faculty of Medicine, University of Ottawa, Ottawa, Canada, **2** University of Ottawa Centre for Infection, Immunity, and Inflammation (CI3), University of Ottawa, Ottawa, Canada

* Subash.sad@uottawa.ca

**Data Availability Statement:** The authors declare that all other data supporting the findings of this study are available within the article and its supplementary information files.

## Abstract

XIAP is an endogenous inhibitor of cell death and inactivating mutations of XIAP are responsible for X-linked lymphoproliferative disease (XLP-2) and primary immunodeficiency, but the mechanism(s) behind these contradictory outcomes have been unclear. We report that during infection of macrophages and dendritic cells with various intracellular bacteria, XIAP restricts cell death and secretion of IL-1β but promotes increased activation of NFκB and JNK which results in elevated secretion of IL-6 and IL-10. Poor secretion of IL-6 by *Xiap*-deficient antigen presenting cells leads to poor expansion of recently activated CD8 T cells during the priming phase of the response. On the other hand, *Xiap*-deficient CD8 T cells displayed increased proliferation and effector function during the priming phase but underwent enhanced contraction subsequently. *Xiap*-deficient CD8 T cells underwent skewed differentiation towards short lived effectors which resulted in poor generation of memory. Consequently *Xiap*-deficient CD8 T cells failed to provide effective control of bacterial infection during re-challenge. These results reveal the temporal impact of XIAP in promoting the fitness of activated CD8 T cells through cell extrinsic and intrinsic mechanisms and provide a mechanistic explanation of the phenotype observed in XLP-2 patients.

## Author summary

Death of host cells is an important physiological mechanism to promote health of an organism, however overt death of host cells can lead to tissue toxicity. X-linked inhibitor of apoptosis (XIAP) is an endogenous inhibitor of cell death that inhibits various enzymes (caspases) that are involved in cell death. In humans, inactivating mutations of XIAP result in X-linked lymphoproliferative disease-2 (XLP-2) which is characterized by a state of lymphoproliferation and immunodeficiency, but the mechanisms are not fully understood. In this report we have tested the role of XIAP in innate and acquired immune

**Funding:** The work was funded by grants from the Natural Sciences and Engineering Research Council of Canada, Grant # RGPIN-2017-03836 to Dr. Subash Sad. The funders had no role in study design, data collection and analysis, decision to publish, or preparation of the manuscript.

**Competing interests:** None of the authors declare any competing interests related to this manuscript

response against two intracellular bacteria and show that XIAP promotes the development of acquired immune response through separate effects on the innate and acquired immune system. While XIAP promotes the expression of IL-6 by cells of the innate immune system which promotes the survival of activated T cells, XIAP also inhibits the culling of activated T cells through a cell intrinsic mechanism. These results provide a new insight into the mechanisms that lead to the development of XLP-2.

## Introduction

An effective primary T cell response to an intracellular pathogen consists of two distinct phases: 1) potent antigen presentation that results in expansion of antigen specific T cells and 2) contraction of the activated T cells via apoptosis [1]. Engagement of TCR and ligands for co-stimulatory molecules by the antigen-presenting cells (APCs) leads to activation and proliferation of T cells. Additionally, APCs secrete cytokines to support T cell survival, acquisition of effector function and memory formation [2]. Immediately after the generation of peak T cell response, usually at day 7 post-infection, a rapid phase of T cell contraction ensues wherein >90% of activated cells are culled [1,3]. Most of these activated cells undergo apoptosis, leaving behind a small population of memory precursors which differentiate to memory cells [4]. Cell survival signals are dominant during the expansion phase of the response to allow >1000-fold expansion in the numbers of specific T cells which differentiate to effectors. Conversely, cell death/apoptotic signals are dominant during the contraction phase of the response, which are responsible for culling of activated cells and return to homeostasis. Thus, cell death and survival signaling is inherently linked to T cell differentiation and memory development [5,6].

Apoptosis is a fundamental mechanism that ensures homeostasis of the immune system. Apoptotic cell death is driven by the activation of the cysteine protease family, known as caspases, which can be activated by the cell intrinsic or extrinsic pathways [7,8]. Caspases are activated during the latter part of the expansion phase of the CD8 T cell response [9]. Support for apoptosis as the mechanism of contraction of CD8 T cell response came from studies which showed decreased contraction of the response in the absence of the pro-apoptotic mediator BIM [6,10]. However, inactivation of apoptotic pathways does not result in zero-contraction of CD8 T cell response. For example, CD8 T cells deficient in *Fasl* undergo normal contraction [11]. Apoptosis can proceed through multiple pathways, and cell death can also occur through non-apoptotic pathways in activated T cells [12,13]. More recently, cell autonomous caspase-8 was shown to restrict the proliferation of CD8 T cells in response to viral infection without any impact on the contraction of the immune response [14].

Inhibitor of apoptosis proteins (IAP) is a family of endogenous inhibitors of caspases and regulators of cell death [15]. Cancer cells often express high levels of IAPs to evade cell death mechanisms [16–18]. Among all the members of the IAP family, X-linking inhibitor of apoptosis protein (XIAP) is the most potent inhibitor of caspase 3, 7 and 9 [19,20]. In the innate immune compartment, XIAP has been shown to promote the activation of NFκB, and various MAPK pathways [21,22]. Due to the pleiotropic functions of XIAP in inducing pro-inflammatory signaling and preventing cell death in innate immune cells, XIAP-deficient mice succumb to infection [21,23,24]. While *Xiap*-deficient T cells stimulated *in vitro* are more prone to apoptosis [25,26], the role of XIAP in T cell differentiation and memory development has been less clear.

Since the contraction of CD8 T cell response occurs despite the presence of the endogenous inhibitors of apoptosis proteins, we hypothesized that the contraction of the response may be

even more severe if the endogenous inhibitors of IAPs were disabled. Here we document the role of XIAP in T cell expansion, differentiation, and memory development. We generated *Xiap−/−* OT1 TCR transgenic mice and evaluated the role of XIAP in the expansion and contraction of CD8 T cell response during infection with *Salmonella enterica* serovar Typhimurium (ST) and *Listeria monocytogenes* (LM). We show that XIAP promotes the expression of IL-6 by antigen-presenting cells to promote the expansion of CD8 T-cell response. In contrast the expression of XIAP in a CD8 T cells modulates their differentiation program that favors the generation of memory cells.

## Results

### XIAP regulates the CD8 T cell response against intracellular bacteria

Since XIAP is a potent endogenous inhibitor of cell death [19,20] and ST induces potent cell death of infected cells [27,28] we tested whether XIAP has any impact on host susceptibility during infection with ST. *Xiap−/−* mice displayed a slightly increased susceptibility against ST (**Fig 1A**). Surprisingly, the bacterial burden was slightly reduced in *Xiap−/−* mice in comparison to WT mice (**Fig 1B**). The relative numbers of various immune cell subsets were not modulated by XIAP in naïve (**S1A Fig**) or infected (**S1B Fig**) mice. Since inflammatory cytokines play important roles in innate immunity [27–29], we measured the expression of inflammatory cytokines in the serum of infected mice at day 5 post-infection and observed that in contrast to WT mice the levels of IL-1β were higher but IL-6 were lower in the serum of *Xiap−/−* mice (**Fig 1C**). To monitor the impact of XIAP on CD8 T cell response we infected mice with recombinant ST that expresses Ovalbumin (OVA). We have previously reported that the expression and cytoplasmic transport of immunogenic proteins, such as Ovalbumin (OVA), through the type III secretion system of ST results in the induction of a potent CD8 T cell response against the protein which promotes the control of infection [30]. We infected mice with ST-OVA ($10^3$ CFU, iv) and observed that the bacterial burden in the spleens of WT and *Xiap−/−* mice was similar during the early and later stages of infection (**Fig 1D**). Interestingly, the CD8 T cell response observed at the peak of response (day 7) was significantly reduced in *Xiap−/−* mice as detected by OVA-dextramer staining (**Fig 1E, 1F**) and by ELISPOT assay (**Fig 1G, 1H**). The decline in CD8 T cell response during the contraction phase was more pronounced in *Xiap−/−* mice (**Fig 1H, 1I**). We also measured the impact of XIAP on CD8 T cell response during infection with OVA-expressing *Listeria monocytogenes* (LM-OVA) [31]. At the peak of infection (day 3) the bacterial burden was substantially higher in *Xiap−/−* mice in comparison to WT mice (**Fig 1J**), From day 5 onwards the bacterial burden was undetectable in both WT and *Xiap−/−* mice. The CD8 T cell response was significantly reduced in *Xiap−/−* mice (**Fig 1K**), similar to what was observed in the ST-OVA infection model. Taken together, these results indicate that XIAP promotes the development of CD8 T cell response independently of the impact on bacterial burden.

### XIAP regulates the expansion of CD8 T cell response through a cell extrinsic mechanism

Since the CD8 T cell response was significantly reduced in *Xiap−/−* mice before the onset of contraction phase (day 8 onwards) we evaluated whether the expression of XIAP in APCs might promote the expansion of CD8 T cell response. We transferred WT OT1 (CD45.1$^+$CD45.2$^+$) CD8 T cells into WT (CD45.1−CD45.2$^+$) and *Xiap−/−* (CD45.1$^+$CD45.2−) mice to monitor their fate before and after infection (**Fig 2A**). Transfer of OT1 CD8 T cells ($10^6$/mouse) into naïve WT and *Xiap−/−* mice resulted in similar numbers of OT1 CD8 T cells

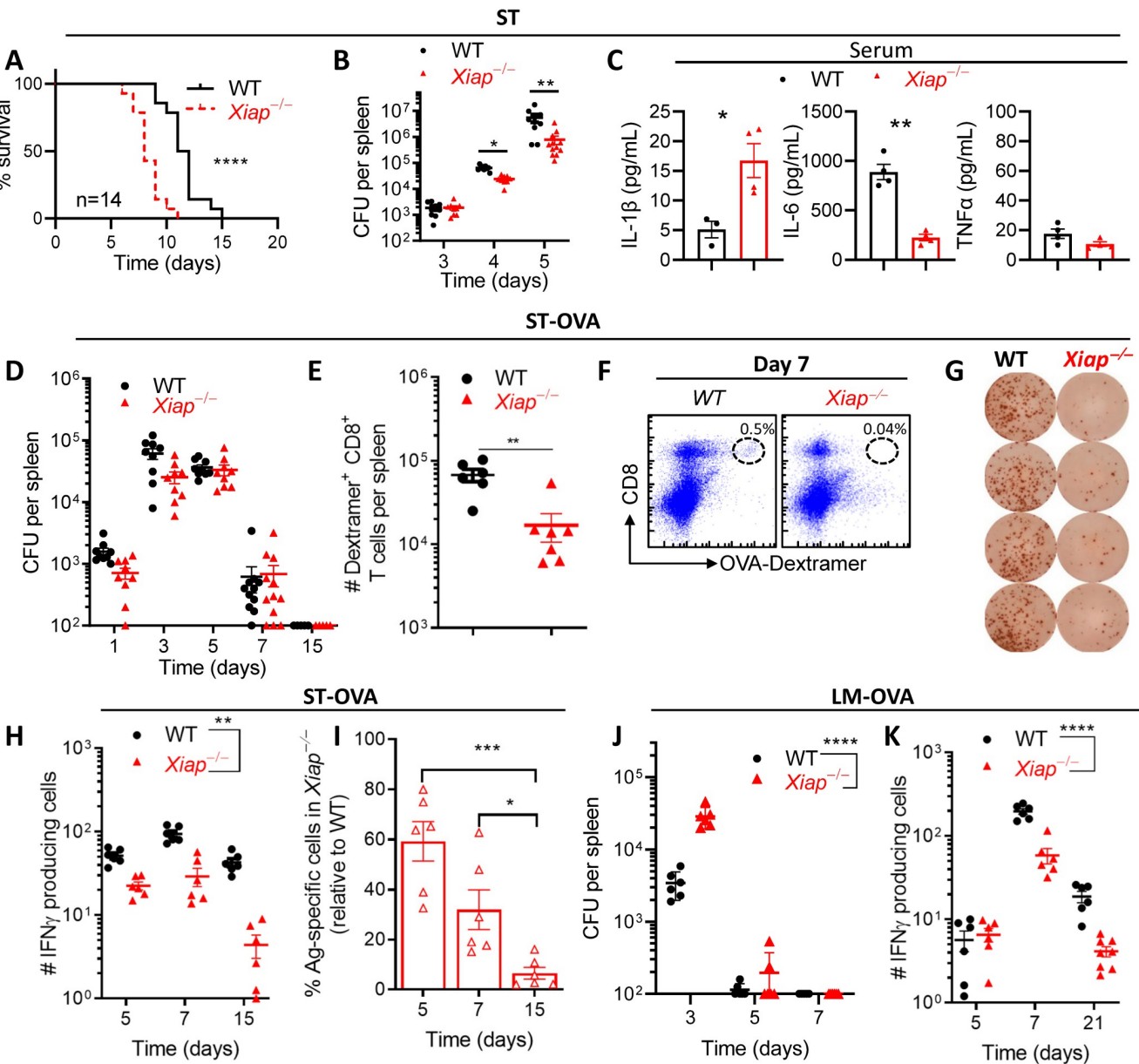

**Fig 1. Poor CD8 T cell response to intracellular bacterial pathogens in *Xiap−/−* mice.** (A-C) WT and *Xiap−/−* mice were infected with ST (200 CFU, i.v.) and the impact on host survival was monitored (**A**). Bacterial burden was measured in the spleens of infected mice at days 3, 4 and 5 post infection (**B**). Cytokine levels were measured in the serum at day 5 post infection (**C**). (**D-I**) WT and *Xiap−/−* mice were infected with ST-OVA ($10^3$ CFU, i.v.) and the spleens were harvested from the infected mice at various timepoints post-infection to evaluate the antigen specific CD8 T cell response to $OVA_{257-264}$ peptide. **D**) Bacterial burden in the spleens. **E**) Numbers and **F**) percentage of $OVA_{257-264}$ (SIINFEKL)-specific CD8 T cells, on day 7 post-infection were evaluated in the spleens of infected mice by staining with anti-CD8 antibody and $H2-K^b-OVA_{257-264}$ Dextramer. (**G-I**) ELISPOT assay was performed in spleen cells stimulated with the $OVA_{257-264}$ peptide *in vitro*. **G**) Representative IFN-γ positive spots in an ELISPOT assay plate. **H**) Number of $OVA_{257-264}$ specific cells secreting IFN-γ in response to $OVA_{257-264}$. **I**) Relative number of antigen specific cells in *Xiap−/−* mice in comparison to the number in WT mice. (**J, K**) WT and *Xiap−/−* mice were infected with LM-OVA ($10^3$ CFU, i.v.). **J**) Bacterial burden in the spleens of infected mice at various time intervals. **K**) Number of $OVA_{257-264}$ specific cells secreting IFN-γ, in response to $OVA_{257-264}$ peptide, evaluated by ELISPOT assay. Data is representative of 2 (A-C, J, K) or 3 (D-I) experiments. Each data point (**B-E, H-K**) represents a separate mouse. Statistical analysis was performed by log-rank test (**A**), unpaired student *t*-test (**B, C, E, I**), and 2-way ANOVA (H, J, K). (*$P<0.05$, **$P<0.01$, ***$P<0.001$, ****$P<0.0001$).

at day 2 post cell transfer (**Fig 2B**). To monitor the fate of transferred OT1 cells following infection with ST-OVA we transferred $5x10^4$ OT1 CD8 T cells per mouse. At day 7 post-infection with ST-OVA the expansion of WT OT1 cells was poor in *Xiap−/−* mice in comparison to

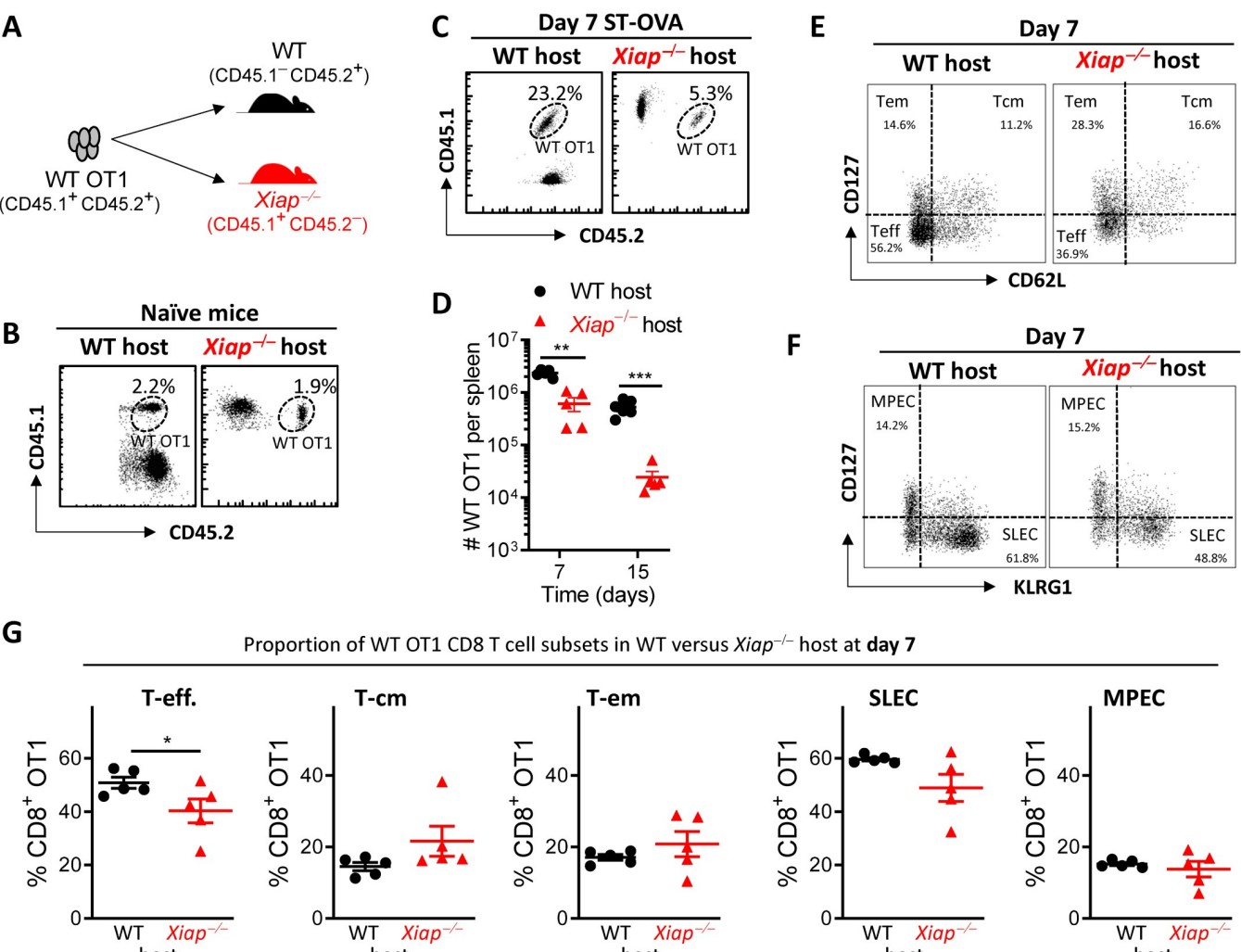

**Fig 2. WT CD8 T cells undergo poor expansion in infected *Xiap−/−* mice. A**) Schematic representation of the adoptive transfer protocol. **B**) CD8 T cells from WT OT1 mice (CD45.1+CD45.2+) were injected (10^6 cells, i.v.) into naïve WT mice (CD45.1- CD45.2+) or *Xiap−/−* mice (CD45.1+ CD45.2-) and the proportion of transferred cells evaluated at day 2 post cell transfer by flow cytometry. (**C-G**) Splenocytes from WT OT1 mice (CD45.1+CD45.2+) were injected (5 x 10^4 cells, i.v.) into WT mice (CD45.1- CD45.2+) or *Xiap−/−* mice (CD45.1+ CD45.2-). Two days later, the recipient mice were infected with ST-OVA (10^3 i.v.). On day 7 and 15 post-infection, the spleens of the recipient mice were harvested, and the donor OT1 cells were tracked by flow cytometry using antibodies against CD8, CD45.1 and CD45.2. (**C**) Representative dot plots and (**D**) numbers of the WT OT1 CD8 T cells in the recipient mice. (**E, F**) Representative dot plots and (**G**) percent distribution of various CD8 T cell subsets within the transferred OT1 populations after staining with various antibodies described in the methods. Data is representative of 3 (**A-D**) or 2 (**E-G**) experiments. Each data point (**D, G**) represents a separate mouse. Statistical analysis was performed by unpaired student *t*-test (*P<0.05, **P<0.01, ****P<0.0001).

WT mice (**Fig 2C, 2D**). The differentiation program of CD8 T cells did not appear to be significantly altered by CD8 T cell extrinsic XIAP except that the proportion of effector phenotype CD8 T cells was slightly modulated (**Fig 2E–2G**). These results suggest that cell extrinsic XIAP promotes the expansion of CD8 T cell response during the priming phase.

## Deficiency of XIAP in APCs results in increased expansion and death of activated CD8 T cells

Since we observed that the poor expansion of CD8 T cell response in *Xiap−/−* mice was due to cell extrinsic role of *Xiap* (**Fig 2**) we tested whether *Xiap−/−* DCs are intrinsically poor at presenting antigen to OT-1 CD8 T cells. Both WT and *Xiap−/−* macrophages and DCs expressed similar

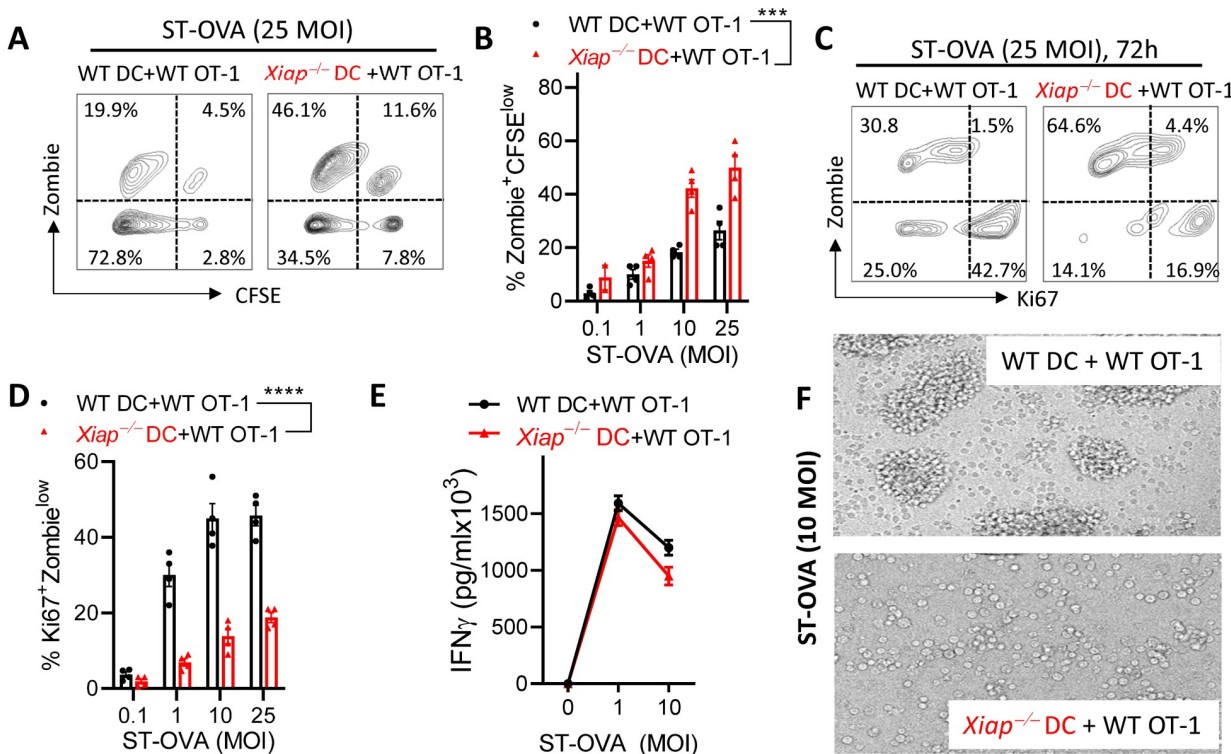

**Fig 3. *Xiap−/−* DCs induce poor antigen presentation to WT CD8 T cells *in vitro*.** (A-F) Bone marrow derived DCs generated from WT and *Xiap−/−* mice were infected with ST-OVA as described in the methods section. Infected DCs were incubated with CFSE labeled purified WT OT1 CD8 T cells. At various time intervals, cells were harvested stained with Zombie Yellow, and antibodies against Ki67 and CD8 and evaluated for proliferation and viability via flow cytometry (**A-D**). Secretion of IFN-γ was evaluated by ELISA in cell supernatants collected at 72h (**E**). Cells were imaged at 72h post culture (**F**). Data is representative of 3 (A-D) or 2 (E, F) experiments. Each data point (**B, D**) represents a separate mouse. Statistical analysis was performed by 2-way ANOVA. (*$P<0.05$, **$P<0.01$).

levels of MHC-I (**S1A Fig**). We infected WT and *Xiap−/−* DCs with ST-OVA and incubated them with CFSE labeled WT OT1 CD8 T cells. We observed that the WT OT1 CD8 T cells that were incubated with the *Xiap−/−* DCs displayed enhanced cycling (CFSE^low or Ki67^hi), however, the primed WT OT1 CD8 T cells also displayed enhanced cell death (Zombie^hi cells) (**Fig 3A–3D**). When WT OT1 CD8 T cells were incubated with ST-OVA infected macrophages, we observed that *Xiap−/−* macrophages promoted increased cycling of WT OT1 CD8 T cells initially but there were reduced numbers of primed WT OT1 CD8 T cells that had completed > 5 cell divisions (**S2B, S2C Fig**). The secretion of IFN-γ by the activated WT OT1 CD8 T cells was not significantly altered whether they were stimulated with WT or *Xiap−/−* DCs (**Fig 3E**) or macrophages (**S2D Fig**). Activated WT OT1 CD8 T cells displayed a "clustering morphology" upon co-culture with WT DCs which occurs during the earlier stage of activation (**Fig 3F**). On the other hand, CD8 T cells activated by *Xiap−/−* DCs had already moved out of activation clusters, displaying later stages of proliferation (**Fig 3F**). We also performed the *in vitro* antigen presentation experiments in the LM-OVA infection model, and here again the WT OT1 CD8 T cells displayed increased proliferation (CFSE^low) and cell death (Zombie^hi) (**S2E–S2G Fig**).

## Cell extrinsic IL-6 partially rescues the poor priming of CD8 T cell response in Xiap−/− mice

Since cell extrinsic XIAP promoted the survival of activated CD8 T cells we wished to determine the mechanism(s) responsible. We first measured the impact on the cytokine production

by antigen-presenting cells. To measure the expression of cytokines cells were infected with ST-OVA that was grown in stationary phase to avoid poor cytokine expression due to premature cell death. The expression of IL-6, and IL-10 was highly reduced in *Xiap−/−* DCs during infection with ST-OVA (**Fig 4A**). The expression of TNFα or IL-12 did not appear to be modulated by XIAP. Similar results were obtained when DCs were infected with LM-OVA (**S3A Fig**) or when macrophages were infected with ST (**S3B Fig**). Since XIAP is an endogenous inhibitor of cell death we measured whether cell death of infected cells was modulated by XIAP. Infection of macrophages with log-phase ST did not reveal any impact of XIAP on cell death (**S3C Fig**). On the other hand, *Xiap−/−* BMMs underwent significantly increased cell death in comparison to WT cells upon infection with ST grown under stationary-phase (**S3D Fig**). Similar results were obtained with DCs (**Fig 4B**). IL-1β which is released following inflammasome activation was also augmented in *Xiap−/−* DCs (**Fig 4B**) or BMMs (**S3D Fig**). Since NLRP3 inflammasome signaling promotes pyroptosis and IL-1β secretion and XIAP has been shown to inhibit NLRP3 [27,32,33] we tested the impact of inhibition of NLRP3 on cell death and IL-1β secretion by *Xiap−/−* macrophages. Inhibition of NLRP3 (MCC950) (**S3D Fig**), RipK1 (Nec-1) (**S3E Fig**) or RipK2 (MDK19922) (**S3F Fig**) did not have any impact on cell death of *Xiap−/−* BMMs. On the other hand, inhibition of RipK2 completely reduced the secretion of IL-1β by *Xiap−/−* BMMs to WT levels (**S3F Fig**) whereas inhibition of NLRP3 or RipK1 had a partial impact. In contrast to ST, *Xiap−/−* DCs did not display increased cell death in response to infection with LM and the secretion of IL-1b was not detectable (**S3G Fig**). These results indicate that the regulation of cell death by XIAP is pathogen dependent.

Correlating with reduced expression of cytokines western blot analysis of cell extracts revealed poor activation of NFκB and JNK in *Xiap−/−* cells (**S4A Fig**). Activation of the MK2, the downstream member of the MAP Kinase pathway was not modulated by XIAP. Correlating with increased cell death, *Xiap−/−* cells displayed increased activation of various caspases (**S4B Fig**), but there was no detectable modulation of gasdermin D cleavage in macrophages (**S4C Fig**) or DCs (**S4D Fig**) implying that the enhanced cell death of *Xiap−/−* cells may not be due to pyroptosis.

We considered the possibility that the enhanced cell death of *Xiap−/−* APCs might be responsible for reduced priming of CD8 T cells, however, we did not observe any impact of XIAP on cell death in response to infection with LM-OVA (**S3G Fig**). Furthermore, inhibition of caspase 1 had no significant impact on the proliferation or death of WT OT-1 cells (**S4F Fig**).

We next considered the possibility that the poor secretion of IL-6 by *Xiap−/−* DCs may lead to reduced proliferation/survival of activated CD8 T cells during the priming phase of response. We therefore added exogenous IL-6 to *in vitro* infected WT and *Xiap−/−* DCs which were incubated with WT OT1 CD8 T cells. The addition of exogenous IL-6 significantly increased the proliferation and survival of WT OT1 cells incubated with infected *Xiap−/−* DCs (**Fig 4C, 4D**). Injection of recombinant IL-6 to ST-OVA-infected *Xiap−/−* mice significantly augmented the numbers of adoptively transferred WT OT1 cells (**Fig 4E–4G**). Taken together, these results indicate that the poor priming of WT CD8 T cells in *Xiap−/−* mice occurs due to poor expression of IL-6 by APCs.

## XIAP regulates the contraction of CD8 T cell response through a cell intrinsic mechanism

Since we observed increased contraction of CD8 T cell response in *Xiap−/−* mice (**Fig 1I**) we wished to evaluate whether this occurred through a cell intrinsic mechanism. We crossbred *Xiap−/−* mice with OT1 mice to generate WT OT1 (CD45.1⁻) and *Xiap−/−* OT1 (CD45.1⁺) mice and adoptively transferred the CD8 T cells of the two genotypes together (1:1) into B6.

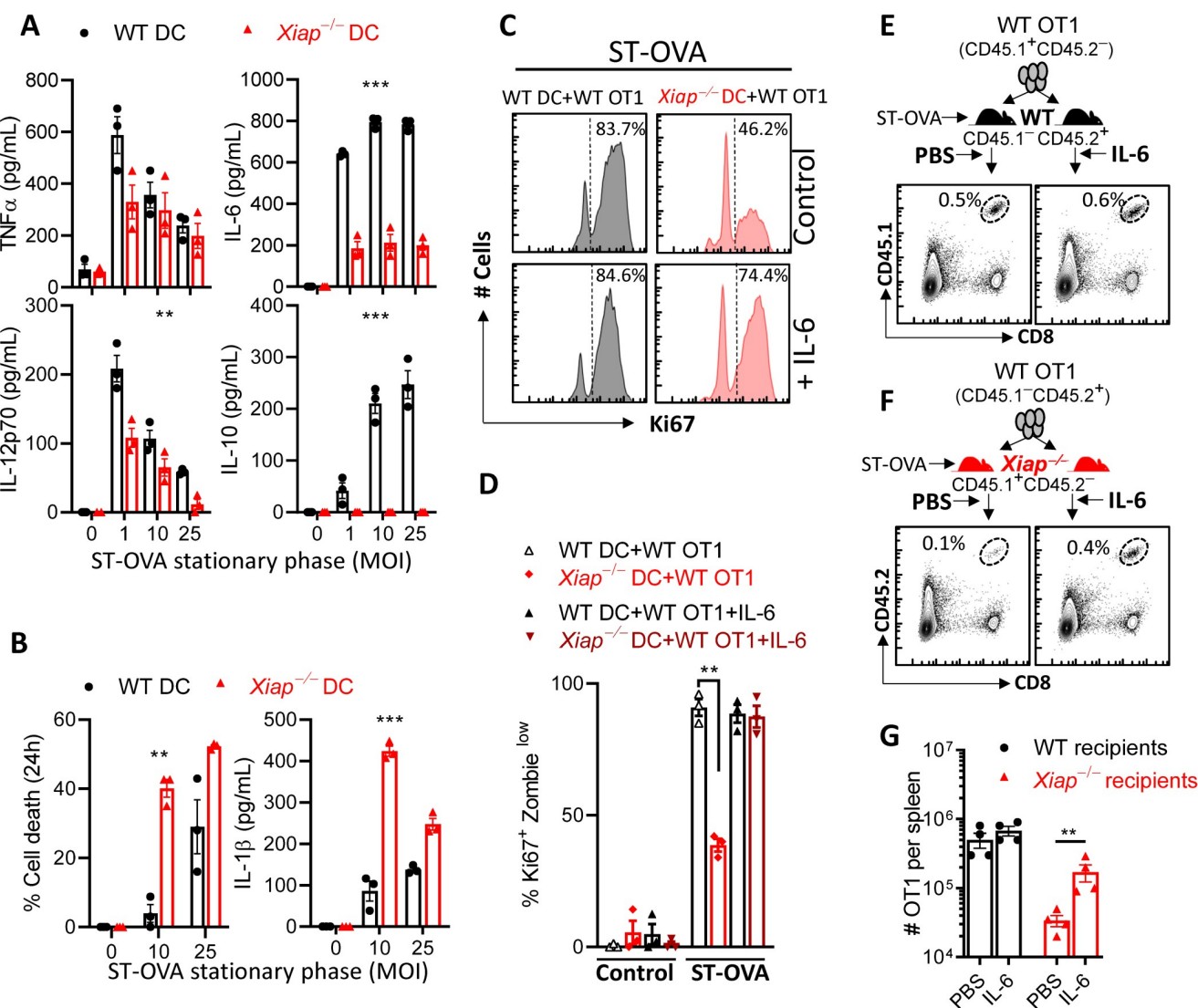

**Fig 4. Impaired priming of CD8 T cells by *Xiap−/−* DCs is due to poor IL-6 expression.** DCs were generated from WT and *Xiap−/−* mice and infected with ST-OVA (**A, B**). Secretion of various cytokines by infected DCs was measured by ELISA in supernatants collected at 24hr post infection (**A**). Cell death of DCs was measured at 24hr post-infection by neutral red assay, and IL-1β secretion was measured by ELISA in the supernatants collected at 24h post infection (**B**). (**C, D**) DCs were infected with ST-OVA (25 MOI) and incubated with purified WT OT1 CD8 T cells in the presence or absence of IL-6 (50 ng/ml). Cell proliferation was measured by staining with Zombie Yellow, and antibodies against Ki67 and CD8 at 96hr. (**E-G**) WT OT1 CD8 T cells were injected ($10^3$/mouse) iv into WT (**E, G**) or *Xiap−/−* (**F, G**) recipient mice. After 24h recipient mice were infected with ST-OVA ($10^3$, ip). IL-6 was injected in several groups of mice (1 μg/mouse) ip on day 1, 3 and 5. Control mice received PBS. On day 7 post infection, spleens were removed from infected mice and spleen cells stained with labeled antibodies against CD45.1, CD45.2 and CD8. Stained cells were acquired on Flow cytometer and the relative numbers of adoptively transferred OT1 CD8 T cells evaluated. Data is representative of 3 (**A-D**) or 2 (**E-G**) experiments. Each data point (**A, B, D, G**) represents a separate mouse. Statistical analysis was performed by 2-way ANOVA (**F**) and unpaired student *t*-test (*$P<0.05$, **$P<0.01$, ***$P<0.001$).

SJL hosts (**Fig 5A**). At day 7 post-infection with ST-OVA, the proportion of WT OT1 versus *Xiap−/−* OT1 cells was similar (**Fig 5B**). This contrasts with the results obtained when WT OT1 cells were transferred to WT versus *Xiap−/−* mice (**Fig 2C, 2D**). This suggests that XIAP does not impact the expansion phase of CD8 T cell response in a cell intrinsic manner. Interestingly, at day 15 post-infection, the decline in *Xiap−/−* OT1 cells was significantly greater than that of WT OT1 cells (**Fig 5B, 5C**). Thus, cell intrinsic XIAP restricts the contraction of CD8 T cell response. We performed phenotypic characterization of WT versus *Xiap−/−* OT1 in the same host at day 7 post-infection (**Fig 5D**) and observed that the XIAP promoted CD8 T cell

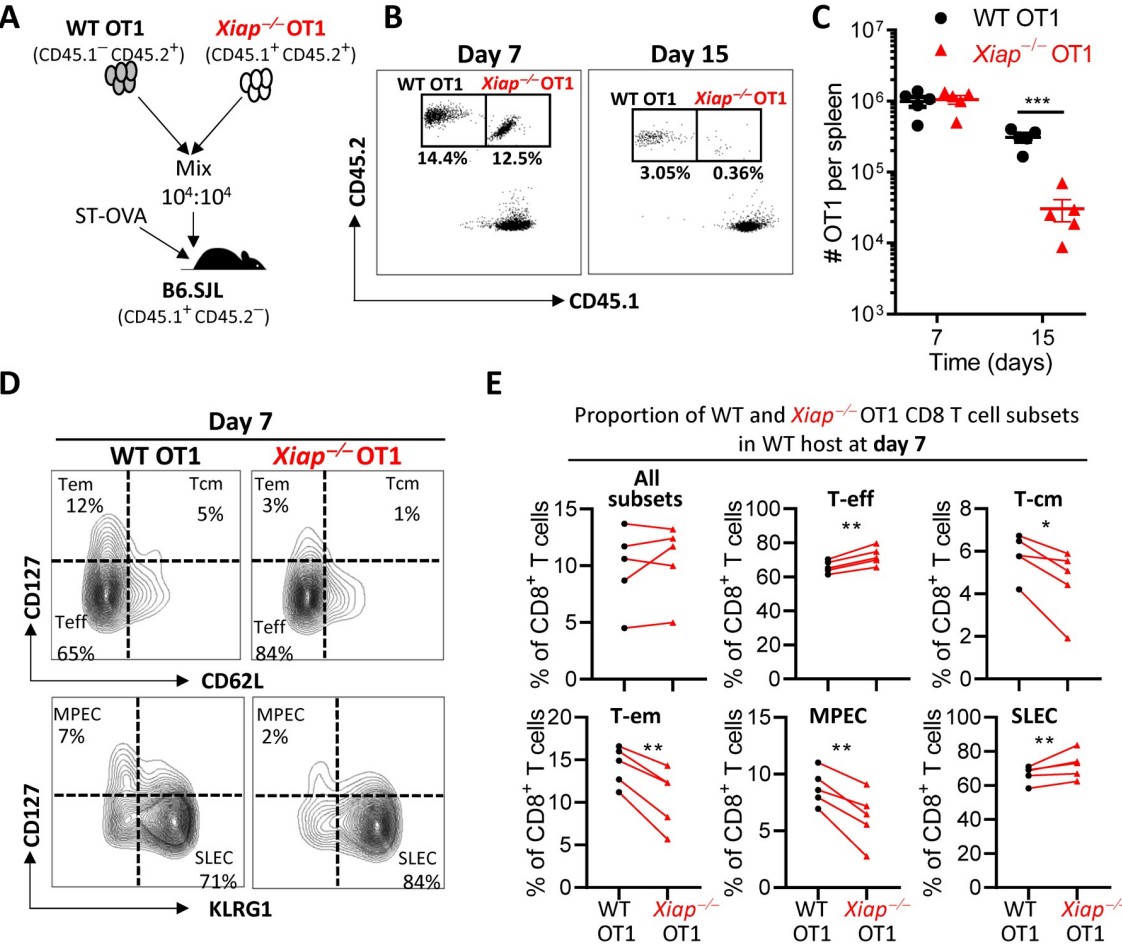

**Fig 5. *Xiap−/−* CD8 T cells undergo increased contraction in WT mice.** (**A**) Schematic representation of the adoptive transfer protocol. Splenocytes from both WT OT1 (CD45.1⁻CD45.2⁺) and *Xiap−/−* OT1 (CD45.1⁺CD45.2⁺) were mixed 1:1 and injected (10⁴ cells each, i.v.) into B6.SJL mice (CD45.1⁺ CD45.2⁻). After two days the recipient B6.SJL mice were infected with ST-OVA (10³ i.v.). At day 7 and 15 post-infection, the spleens of the recipient mice were harvested, and the donor OT1 cell numbers evaluated by flow cytometry using antibodies against CD8, CD45.1 and CD45.2. Representative dot plots (**B**), and cell number (**C**) of WT and *Xiap−/−* OT1 CD8⁺ T cells in the WT recipient mouse is shown. Representative dot plots (**D**) and the distribution (**E**) of various OT1 CD8⁺ T cell subsets in the same host based on flow cytometric analysis following staining with antibodies is shown. Data is representative of 3 (**A-C**) or 2 (**D, E**) experiments. Each data point (**C, E**) represents a separate mouse. Statistical analysis was performed by unpaired student *t*-test (**C**) (****P<0.0001), and 2-way ANOVA (**E**).

differentiation into the memory precursors (MPEC) and central memory (T-cm) phenotype (**Fig 5E**).

The experiments described above involved the transfer of WT and *Xiap−/−* OT1 cells into a WT host. We also performed this adoptive transfer of WT and *Xiap−/−* OT1 cells into *Xiap −/−* recipients (**S5A Fig**). Two key observations were made. 1) There was a slight increase in the expansion of *Xiap−/−* OT1 in comparison to WT OT1 (**S5B, S5C Fig**). 2) The overall expansion of the OT1 cells (at day 7) was much reduced when the cells were transferred into *Xiap−/−* recipients (**S5B Fig**) in comparison to when the cells were transferred to WT hosts (**Fig 5B**), which could be due to poor IL-6 expression by *Xiap−/−* APCs. In both WT (**Fig 5D, 5E**) and *Xiap−/−* (**S5C Fig**) hosts, we observed that a smaller proportion of *Xiap−/−* OT1 CD8 T cells differentiated to the memory phenotype, compared to WT OT1 cells. Increased proportion of the *Xiap−/−* OT1 retained effector phenotype, suggesting a potential defect in memory formation and increased effector T cell activity.

## CD8 T cell intrinsic XIAP restricts cycling and promotes the survival of activated cells

Having observed increased contraction of *Xiap−/−* OT1 cells in comparison to WT OT1 cells we wished to understand the mechanism involved in the *in vitro* cell culture model. We infected WT DCs with ST-OVA *in vitro* and incubated them with CFSE labeled WT OT1 or *Xiap−/−* OT1 cells and observed a slight increase in the numbers of proliferating *Xiap−/−* OT1 cells in comparison to WT OT1 cells (**Fig 6A**). *Xiap−/−* OT1 cells displayed increased proliferation (Ki67$^+$) (**Fig 6B**), and the numbers of live, cycling *Xiap−/−* OT1 cells were increased in comparison to WT OT1 cells at early time intervals (**Fig 6C**). Similar results were obtained when DCs were infected with LM-OVA (**S6A–S6D Fig**). Interestingly, the expression of cleaved caspase-3 was increased in *Xiap−/−* OT1 cells in both infection models (**Figs 6D, 6E** and **S6E**). *Xiap−/−* OT1 cells displayed increased effector activity by secreting enhanced levels of IFN-γ in comparison to WT OT1 cells upon incubation with WT DCs infected with ST-OVA- (**Fig 6F**) or LM-OVA- (**S6F Fig**). Interestingly, *Xiap−/−* OT1 cells expressed reduced levels of IL-2 (**Fig 6G**), which is indicative of increased differentiation towards effector cells with a compromised potential to proliferate long-term. The activation of NFκB was reduced (**Fig 6H**) and the activation of caspase-8, 9 and 3 was modestly increased in *Xiap−/−* OT1 cells (**Fig 6I**), indicative of poor survival potential. When the *in vitro* stimulated OT-1 cells were cultured for longer time intervals *in vitro* we observed poor survival of *Xiap−/−* OT1 cells (**Fig 6J**). Taken together, these results indicate that cell intrinsic XIAP promotes increased differentiation of CD8 T cells towards memory precursors which results in enhanced generation of memory cells.

## Xiap-deficient CD8 T cells generate poor memory response

A key aspect of memory cells is their ability to persist long-term and mediate a rapid recall response following a re-exposure to the same antigen/pathogen. Having observed increased contraction of *Xiap−/−* OT1 cells in comparison to WT OT1 cells we wished to test the long-term survival of WT versus *Xiap−/−* OT1 CD8 T cells in infected mice. We adoptively transferred WT versus *Xiap−/−* OT1 CD8 T cells into WT hosts which were infected with ST-OVA (**Fig 7A**). At day 30 post-infection we observed that the proportion of *Xiap−/−* OT1 CD8 T cells was reduced in comparison to WT OT1 cells in the same WT host (**Fig 7B**). Following a challenge with LM-OVA we observed that there was a significantly higher expansion in the numbers of WT OT1 CD8 T cells in comparison to *Xiap−/−* OT1 CD8 T cells (**Fig 7B, 7C**). We also tested whether memory CD8 T cells in *Xiap−/−* mice mediate effective control of ST-OVA following a re-challenge (**Fig 7D**). Our results indicate that there is poor control of ST-OVA following a re-challenge with a higher dose of ST-OVA (**Fig 7E**). Taken together, these results indicate that a deficiency of XIAP while promoting cycling and effector function of CD8 T cells initially compromises the ability of primed CD8 T cells to develop into memory cells which results in poor control of infection, thereby creating a state of immunodeficiency.

## Discussion

The endogenous inhibitors of apoptosis proteins (IAPs) inhibit caspase activation [15], however their presence does not appear to deter the massive culling (~90%) of the T cell response during the contraction phase. IAPs restrict cell death by directly inhibiting the enzymatic activity of caspases or by mediating ubiquitin editing of its substrates. XIAP is the only mammalian IAP with the ability to directly inhibit enzymatic activity of two effector caspases, caspase-3 and -7, and the initiator caspase, caspase-9 [19,20,34–37]. It is conceivable that the IAPs are not active in T cells, or the inhibition of caspases by IAPs in activated T cells is not strong

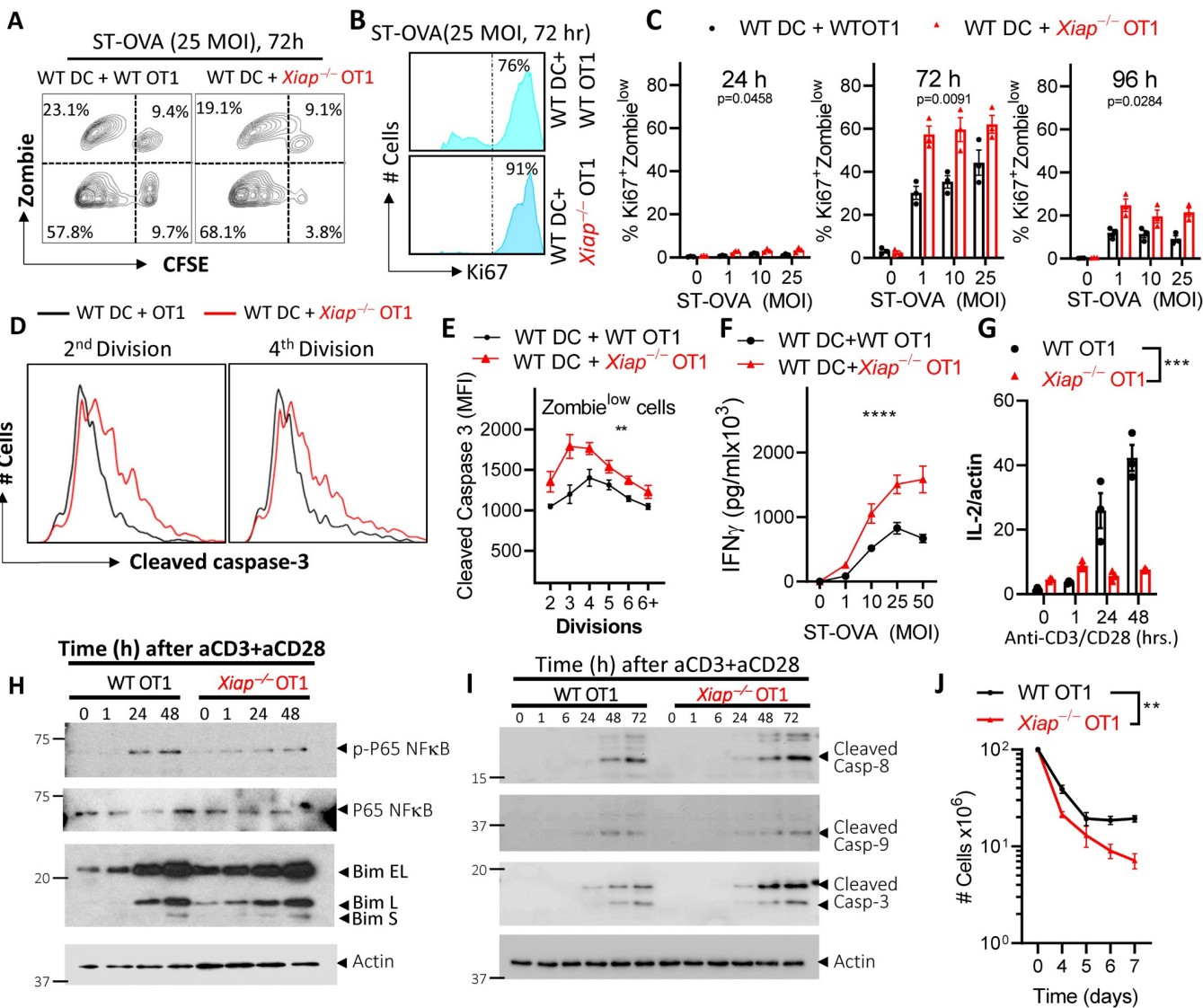

**Fig 6. Cell intrinsic XIAP restricts cell cycling and death of activated CD8 T cells. (A-D)** Bone marrow derived dendritic cells were generated from WT mice and infected with ST-OVA. CD8 T cells were purified from WT OT1 and *Xiap*−/− OT1 spleens and labelled with CFSE and incubated with the infected WT DCs. At the indicated time intervals cells were stained with Zombie Yellow, and antibodies against Ki67, caspase-3 and CD8 and evaluated for proliferation and viability via flow cytometry. (**A**) Representative contour plot showing proliferation and cell death of OT1 CD8 T cells. (**B**) Representative histograms showing Ki67+ of OT1 CD8 T cells. (**C**) Graphs showing Ki67+ Zombie$^{low}$ OT1 CD8 T cells. (**D, E**) Expression of cleaved caspase-3 was evaluated by flow cytometry. (**F**) Supernatants were collected at 24hr post DC+OT1 CD8 co-culture and IFN-γ secretion evaluated by ELISA. (**G-I**) WT and *Xiap*−/− CD8 T cells were stimulated with plate-bound anti-CD3 (1 μg/ml) + anti-CD28 (10 μg/ml) antibodies. Expression of IL-2 was measured by qRT-PCR analysis at various time intervals (**G**). Activation of NFκB and BIM (**H**) and various caspases (**I**) was evaluated at various time intervals by western blotting of cell extracts. (**J**) WT OT1 and *Xiap*−/− OT1 spleen cells (10$^7$/mL) were incubated with ST-OVA (10$^3$) for 3h followed by culture in media containing gentamicin. Cells were diluted 3-fold daily after day 3 with supplementation of fresh IL-7 (1 ng/ml). Cells were harvested at various time intervals and counted. Data is representative of 3 (**A-C, F, G**) or 2 (**D, E, H-J**) experiments. Each data point (**C, G**) represents a separate mouse. Statistical analysis was performed by 2-way ANOVA. (**P<0.01, ***P<001).

enough to prevent the culling of T cell response. However, polyclonal stimulation of *Xiap*$^{-/-}$ T cells *in vitro* has been shown to result in increased apoptosis [25,26], however, the role of XIAP in T-cell differentiation and memory development has not been evaluated. Since XIAP is the *bona fide* endogenous inhibitor of apoptosis we evaluated its role during infection with an acute and a chronic intracellular bacterium. We show that XIAP signaling in macrophages/

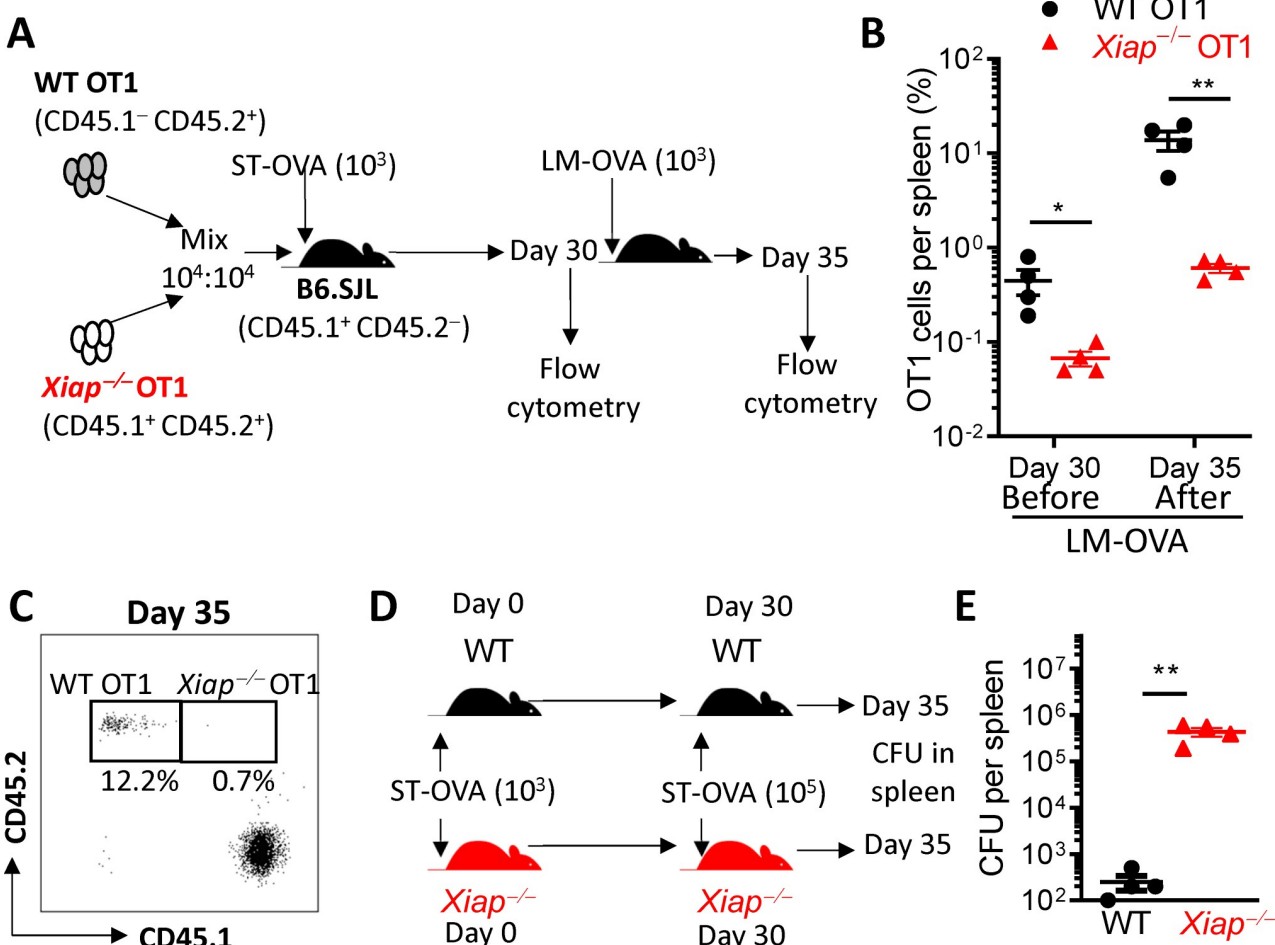

**Fig 7. Cell intrinsic XIAP promotes functional CD8 T cell memory.** (A-C) WT and *Xiap−/−* OT1 cells were adoptively transferred into WT hosts and infected with ST-OVA as described in panel **A**. A group of ST-OVA infected mice were challenged with LM-OVA on day 30. On day 30 and day 35 post ST-OVA infection, the numbers of adoptively transferred OT-1 cells were enumerated in recipient mice. Dot plots (**C**) and the proportion (**B**) of WT and *Xiap−/−* OT1 cells among CD8 T cells are shown. (**D, E**) WT and *Xiap−/−* mice were infected with ST-OVA ($10^3$, iv). On day 30 post-infection mice were re-challenged (**D**) with a higher dose of ST-OVA ($10^5$, iv) and bacterial burden evaluated in the spleens of mice five days later (**E**). Data is representative of 3 (**A, B**) or 2 (**C-E**) experiments. Each data point (**B, E**) represents a separate mouse. Statistical analysis was performed by paired student *t*-test (*$P<0.05$, **$P<0.01$, ***$P<0.001$).

DCs and CD8 T cells promotes the development of T cell memory response through separate mechanisms.

Recent work has indicated that XIAP not only inhibits apoptotic pathways, but it also regulates the 'ripoptosome' and inflammasome complexes [33,38–40]. IAPs interact with the various members of the TNF-R signaling complex such as RipK1 to orchestrate signaling through various downstream cascades. IAPs regulate RipK1 ubiquitination to transition TNFα signaling towards a pro-survival NF-κB pathway [41]. While TNFα induces pro-survival NFκB signaling [42], cotreatment with an inhibitor of NFκB switches cell signaling towards cell death [41,43]. Inhibition of IAPs disengages RipK1 from the pro-survival complex leading to RipK1-RipK3 interaction which promotes inflammasome signaling, cell death and IL-1β secretion through the RipK3-caspase-8-NLRP3 pathway [33,39,40]. XIAP has been shown to inhibit NLRP3 inflammasome signaling [44]. While we observed that inhibition of NLRP3 reduced IL-1β secretion during infection with ST, cell death and IL-1β secretion by *Xiap−/−* cells was still higher than WT cells, suggesting that XIAP inhibits additional cell death pathways to

control cell death and IL-1β secretion by APCs. It has been reported that multiple cell death pathways are engaged during infection of cells with ST [45].

Although IAPs were initially identified as direct inhibitors of caspases, later studies have revealed alternative roles for IAPs as E3-ubiquitin ligases that participate in the regulation of pattern-recognition receptor (PRR) signaling [46–50]. XIAP has been shown to promote NFkB signaling via K63-linked ubiquitination of TAK1 [46, 50, 51]. In contrast to the positive regulation of NFκB by XIAP, the impact of XIAP on JNK activation is not clear as some reports indicate that XIAP inhibits JNKs signaling through K48-linked ubiquitination and proteasomal degradation of TAK1 [52] whereas others have shown that XIAP promotes JNK activation [21,53,54]. The role of IAPs has traditionally been evaluated in tumor cell lines which typically do not express inflammatory cytokines. Modulation of NFκB signaling in myeloid cells by XIAP is bound to influence the expression of inflammatory cytokines. Inactivating mutations of *Xiap* in XLP-2 patients have been shown to result in defective NOD2-mediated NFκB activation [55]. We also observed a reduction in NFκB signaling in infected *Xiap*-deficient macrophages and DCs leading to reduced expression of IL-6 and IL-10. It has been previously reported that LPS+IFN-γ primed *Xiap−/−* macrophages express reduced levels of IL-6 and TNFα upon infection with *Listeria monocytogenes* [21]. Reduction in the levels of pro-inflammatory cytokines was also observed in the serum of *Xiap−/−* mice infected with *Chlamydophila pneumoniae* [23]. Both studies attributed the poor cytokine expression by *Xiap−/−* macrophages to the attenuation of NFκB and JNK signaling. Concurrent with the idea of poor-innate immune response in the context of XIAP deficiency, both studies reported an increased pathogen burden in *Xiap−/−* mice following a pathogen challenge. Interestingly, we observed that the impact of XIAP is pathogen dependent. In *Xiap−/−* mice the control of LM is compromised whereas the control of ST is marginally enhanced. Thus, XIAP appears to have a contrasting impact on the control of *L. monocytogenes* versus S. *typhimurium*. The enhanced cell death signaling in *Xiap−/−* mice may result in reduced ST burden at the cost of excessive inflammation. We did not observe any significant impact of XIAP on the secretion of TNFα by DCs or macrophages. It has been shown that cIAPs and XIAP redundantly inhibit the Ripk1/RipK3-dependent TNFα secretion by macrophages [39,56]. It is therefore likely that cIAPs compensate to promote TNFα secretion by *Xiap−/−* cells.

Expression of cytokines by the infected antigen-presenting cells provides the third signal that ensures increased cycling of stimulated T cells [57]. Type I interferon and/or IL-12 have been shown to provide the third signal for promoting cycling of primed CD8 T cells [58]. In addition to this, some cytokines expressed by APCs such as IL-6 promote the survival of proliferating T cells which results in increased numbers of activated T cells [59–61]. Poor expression of IL-6 by the antigen-presenting cells compromises the priming/survival of T cells [59]. IL-6 rescues T-cells from apoptosis by inhibiting the downregulation of Bcl-2 in a dose dependent manner which promotes the survival of stimulated T-cells [60,61]. There are multiple pathways through which T cells can undergo cell death and deprivation of the key survival cytokines can result in increased death of T cells [62–65]. Inflammatory cytokines such as TNFα have been shown to promote cell death of T cells through TNF-R1 signaling [66]. Finally, activated T cells have been shown to undergo cell death through a process described as "activation-induced cell death (AICD)" *in vitro* [67]. T cells from mice with a defective *Fas* or *FasL* gene were shown to be resistant to AICD *in vitro* [68, 69]. Furthermore, AICD was shown to occur in mature T cells stimulated through Fas-FasL interactions [70–72]. Although there is increased accumulation of T cells in the lymph nodes of mice that are defective in Fas/FasL [73], we have reported that the contraction of CD8 T cell response following infection with LM is not influenced by Fas/FasL interactions [11]. It should be noted that the studies that revealed the role of Fas in AICD involved *in vitro* stimulation of T cells. While the role of Fas-

pathway in the apoptosis of T cells is important, it does not seem to play any role in the contraction of CD8 T cell response during infection. Fas-engagement promotes apoptosis by the extrinsic pathway which is critically dependent on activation of caspase-8 [74], but the T cell intrinsic caspase-8 is required for survival, not death of T cells [75]. Although XIAP has not been traditionally viewed as an inhibitor of caspase-8 our results indicate increased caspase-8 activation in *Xiap*−/− CD8 T cells.

Since TCR-stimulation by APCs results in caspase 3/7 dependent apoptosis [76–79], it is likely that the intrinsic pathway of apoptosis that starts with the apoptosome formation and activation of caspase-9 is mainly involved in the contraction of CD8 T cell response. XIAP inhibits caspase-9 [20] as well as caspase-3/7 [19], suggesting that XIAP might play a major role in the contraction of CD8 T cell response. However, we did not observe any significant impact of XIAP on caspase-9 activation in CD8 T cells. The loss of 90-95% of activated CD8 T cells after priming in WT mice *in vivo* suggests that either XIAP is not relevant in activation-induced cell death of CD8 T cells, or perhaps XIAP is responsible for limiting the extent of this contraction even further. Our results indicate that the contraction of the response was further enhanced in *Xiap*−/− CD8 T cells, suggesting that XIAP does indeed reduce the magnitude of contraction. *Xiap*−/− T cells activated *in vitro* underwent increased cell death [25]. Previously XIAP was shown to have no impact on CD8 T cell response evaluated on day 7/8 of infection with *Lymphocytic Choriomeningitis Virus* (LCMV), and treatment of mice with an inhibitor that blocks all IAPs resulted in reduced numbers of antigen specific CD8 T cells [80]. Our results in both the acute and chronic infection model indicate that during the priming phase, the number of antigen specific CD8 T cells is reduced in *Xiap*−/− mice, and this occurs in a CD8 T cell extrinsic manner. Furthermore, XIAP exerts an even greater impact during the contraction phase through a CD8 T cell intrinsic pathway. It is quite likely that there is compensation by other IAPs which prevent further reduction in antigen specific CD8 T cells in *Xiap*−/− mice. Differentiation of naïve CD8 T cells has been previously reported to proceed through an intermediate state that is promoted by WNT signaling [81–83]. WNT-signaling is disabled in the nucleus by the co-repressors GRO/TLE (Groucho/transducin-like enhancer proteins) which inhibits the formation of the β-catenin-TCF transcriptional activation complex [84,85]. In Xenopus and HEK293 cells it has been previously reported that XIAP monoubiquitinates TLE which disrupts the association of TLE with TCF/Lef, allowing for β-catenin-TCF/Lef assembly and initiation of the WNT transcriptional programs [86,87]. We observed that *Xiap*−/− CD8 T cells expressed poor levels of IL-2 indicating a possible involvement of WNT-signaling in promoting CD8 T cell response.

Inactivating mutations of *Xiap* lead to two contradictory outcomes: lymphoproliferative disease and immunodeficiency [88,89]. We also observed that *Xiap*−/− CD8 T cells displayed increased proliferation and expressed higher levels of IFN-γ. A similar increase in IFN-γ production by T cells was observed in patients with XIAP deficiency [90]. It has been previously reported that caspase activity is required for cycling of recently activated T cells [9,91]. It is therefore conceivable that increased caspase activation in *Xiap*−/− CD8 T cells results in increased cell cycling during the priming phase. It is possible that this is due to cleavage-dependent inhibition of a potential inhibitor of cell cycling protein [92]. We have shown that the enhanced cycling of *Xiap*−/− CD8 T cells during the priming phase was cell intrinsic. This explains the lymphoproliferative phenotype that is observed in XLP-2 patients. Despite the increased proliferation during the initial period, we observed that *Xiap*−/− CD8 T cells undergo enhanced contraction subsequently which results in compromised development of T cell memory. Poor development of memory T cell response due to inactivation of XIAP could result in a state of immunodeficiency. Hence, the over excited T cell response during the initial period followed by poor maintenance of activated T cells in the context of XIAP deficiency could explain the clinical manifestation of XLP-2 patients.

## Methods

### Ethics statement

All the strains of mice were kept at the Animal Facility of the University of Ottawa. Maintenance of mice and the experimental procedures were done in accordance with the guidelines of the Canadian Council of Animal Care (CCAC). The animal use protocol, BMI-3486, was approved by the University of Ottawa Animal Care Committee.

### Mice strains

All mouse strains were bred and maintained in a specific pathogen free facility at the Animal Care and Veterinary Services at Roger Guindon Hall (University of Ottawa, Ottawa, Ontario). Wildtype and *Xiap*−/− mice were housed in the same cage post weaning. All procedures were done in compliance with the Canadian Council on Animal Care guidelines. Wildtype (WT) C57BL/6J (CD45.2$^+$) mice, B6.SJL(CD45.1$^+$) and OT1 TCR transgenic (CD45.2$^+$) mice were obtained from The Jackson Laboratory (Bar Harbor, Maine, USA). *Xiap*−/− mice, described previously [93], were obtained from Dr. Robert Korneluk (Children's Hospital of Eastern Ontario, Canada) and backcrossed into the B6.SJL (CD45.1$^+$ background). *Xiap*−/− OT1 mice were generated by mating OT1 (CD45.2+) mice with *Xiap*−/− (CD45.1+) mice. WT OT1 (CD45.1+ CD45.2+) mice were generated by mating OT1 (CD45.2+) mice with B6.SJL (CD45.1+) mice. All mice used in this study were between the ages of 6-8 weeks. For most of the *in vivo* experiments the breeding strategy involved mating heterozygous breeders and evaluating the response (survival, CFU, CD8 T cell) in +/+ and −/− littermates. In experiments that involved tracking of WT versus *Xiap*−/− OT1 TCR transgenic CD8 T cells a different breeding strategy was adopted wherein the WT and *Xiap*−/− OT1 TCR transgenic CD8 T cells expressed differential CD45 isoforms. In such cases the donor cells were from separate cages. However, the donor WT and *Xiap*−/− mice were kept in the same cage from the third week of birth to equilibrate the microbiota.

### Bacteria and infections

Infections were performed with *Salmonella enterica* serovar Typhimurium (ST, SL1344) and *Listeria monocytogenes* (LM, 10403S) strains, and the recombinant versions of these strains that express Ovalbumin (OVA). Construction of these recombinants has been reported previously [30,94]. Bacteria were resuspended in 100μl PBS and mice were infected via lateral tail vein. For *in vitro* infections, bone marrow derived macrophages (BMDM) and dendritic cells (BMDC) were plated at $10^5$ cells per well in a round flat bottom 96-well plate in RPMI media (Rosswell Park Memorial Institute (RPMI) -1640 + 50μM β-mercaptoethanol) containing 8% fetal bovine serum (FBS) (R8). Actively growing bacteria in LB media (mid log phase) or bacteria grown in LB media overnight (stationary phase) were added to the cells at various multiplicity of infection (MOIs). Plates were centrifuged at 2500 rpm for 6 minutes to synchronize bacterial uptake. Cells were incubated for 30 mins, followed by a two-hour incubation with gentamicin (50 μg/ml) to remove extracellular bacterium. Media was replaced with fresh R8 containing 10 μg/ml gentamicin for various time intervals. Cytokine production and cell death assays were performed at various time intervals.

### Generation of macrophages and dendritic cells

Bone marrow progenitors isolated from the femur and tibia of representative mouse strains were cultured with various growth factors to generate bone marrow derived- dendritic cells (BMDC) and macrophages. Progenitor cells were cultured at a concentration of 1 million

cells/ml in 10 ml R8 medium in the presence of 5 ng/ml GM-CSF (Empire Genomics, Buffalo, NY) in T25 flasks at 37˚C to generate BMDCs [95]. BMDMs were generated by coating petri dishes with 50 ng of M-CSF and plating $10^7$ progenitors in 10 mL R8 medium as per our published procedures [96]. Cells were used at day 7 of differentiation.

## Flow cytometry

Cell surface staining was performed in phosphate buffered saline (PBS) solution containing 1% BSA at 4˚C. Cells, $10^6$ per tube, were treated with anti-mouse Fc block (anti-CD16/32) for 10 minutes followed by surface staining with optimal concentration of the fluorophore-conjugated surface antibodies for 30 minutes. Staining with H2-K$^b$-OVA$_{257-264}$ dextramer conjugated to PE (Immudex, Denmark) was performed for 10 minutes at room temperature, prior to Fc block and surface staining. Intracellular staining was conducted using BioLegend's True-Nuclear Transcription Factor Buffer Set, following manufacture's protocol. Antibodies against the following proteins were obtained from eBioscience (CD8α, #17-0081-82; CD11b, #48-01112-82; CD19, #48-0193-82; CD45.2, #48-0454-80; CD4, 17-0042-82; CD3, #17-0032-82; NK1.1, 17-5941-82; MHC II, #175321-81; CD62L, #48-0621-82; MHC I H2K$^b$, #17-5958-82; CD49b, #17-5970-81; FoxP3, #17-5773-80; RORγT, #17-6981-82). Antibodies against the following proteins were obtained from Invitrogen (Ki67, #46-5698-82; CD127, #17-1271-82; Ly6C, #12-5932-80; Ly6G, # 11-5931-82; CD45.1, #12-0452-82). Anti-cleaved caspase-3 (Asp175) was obtained from Biotechne (5A1E, #IC835R). Antibody against CD11c was obtained from BD Biosciences (#553800). Antibody against KLRG1 was obtained from Biolegend (#138409). Intracellular staining for Ki67 or FoxP3 was performed after treatment of cells with Cytofix/Cytoperm.

## Cell death assays

Zombie Yellow from Biolegend (San Diego, CA) was used for live-dead staining. Zombie is an amine-reactive fluorescent dye that is only permeant to cells with compromised membranes. Cells were washed with PBS and incubated with Zombie Yellow for 30 minutes at 37˚C. Excess dye was washed using 1% BSA in PBS and cell death was measured by flow cytometry. Cell death was also evaluated by the neutral red uptake assay as described previously [32]. 100 μl of neutral red dye was added to the wells at a concentration of 0.17g/mL, in R8 media, and incubated for 5-10 minutes at 37˚C. Excess dye was washed with PBS. Neutral red dye within cells was solubilized using 50% ethanol, and 1% glacial acetic acid solution. Absorbance was measured at 570-650nm using the FilterMax F5 microplate reader from Molecular Devices (Sunnyvale, CA). Percent cell death was calculated relative to absorbance values from control wells.

## ELISPOT assays

Enumeration of IFN-γ-secreting cells was conducted by Enzyme-linked immunospot assays (ELISPOT) on spleens harvested from mice infected with OVA expressing pathogens. Various dilutions of spleen cells were cultured in the presence of IL-2 (0.1ng/mL) with/without peptide (OVA$_{257-264}$, 5μg/mL) for 30 hours, and assays were developed as reported previously [97].

## Cytokine secretion

Supernatants from *in vitro* experiments were collected and analyzed for the presence of cytokines by ELISA (kits obtained from BD Bioscience, San Diego, CA). TMB (3,3',5,5'-tetramethylbenzidine) was used as a chromogenic substrate. Reaction was stopped with 0.2 M sulfuric acid. Absorbance was measured at 450nm using the FilterMax F5 microplate reader

from Molecular Devices (Sunnyvale, CA). Expression of type I interferon (IFN-1) was measured using an L929 cell line (obtained from Dr. B. Beutler) with a luciferase reporter gene cloned under the regulation of an interferon-stimulated response element (ISRE) promoter. ISRE-L929 cells ($5\times10^4$ cells per well in 96-well plates) were incubated with 40 μl cell culture supernatant, for 4 hrs at 37˚C. Luciferase activity was determined using a Luciferase Assay System according to the manufacturer's protocol (Promega).

## Antigen presentation assay

The harvested BMDMs/BMDCs were plated at $5x10^4$ cells per well and infected with ST-OVA or LM-OVA for 1 hour as described above. Following 1h infection with bacteria, APCs were co-cultured with purified (see below) $5x10^4$ OT1 CD8 T cells per well, in R8 medium containing gentamycin (10μg/mL). Proliferation and death of T cells was evaluated at various timepoints by flow cytometry. Exogenous recombinant murine IL-6 (rm IL-6), from Peprotech (Cranbury, NJ) was added at 50 ng/ml. Spleens of OT1 were labelled with Carboxyfluresceine succinimidyl ester (CFSE), obtained from Sigma (St Louis, MO), to assess cell proliferation. Splenocytes were incubated, at a concentration of $1x10^7$ per ml, with 0.125μM CFSE in PBS at 37˚C for 8 min. Excess CFSE was quenched by adding an equal volume of equine serum. Cells were thoroughly washed with PBS and CD8 T cells isolated using EasySep CD8 T cells negative selection kits from Stemcell (Vancouver, British Columbia). At various time intervals cells were harvested, stained with anti-CD8 antibody and the dilution of CFSE stain evaluated by Flow cytometry.

## Adoptive transfer

OT1 mice express a transgenic TCR that is specific for H2-$K^b$-OVA$_{257-264}$. Three days prior to infection, OT1 cells were injected (i.v.) into recipient mice. Spleens from OT1 mice were mechanically disrupted using frosted glass slides, followed by treatment with ACK buffer. Following infection of recipient mice, T cell response was measure by assessing the number of adoptively transferred OT1 CD8 T cells present in the recipient mouse. Splenocytes were harvested as described above and stained for congenic CD45 surface marker (CD45.1 and CD45.2) and assessed by flow cytometry. For cell intrinsic model, $10^4$ of WT-OT1 (expressing CD45.2$^+$ CD45.1$^-$) and $10^4$ of *Xiap*−/− OT1 (expressing CD45.2$^+$ CD45.1$^+$) splenocytes were mixed and injected into B6.SJL (expressing CD45.2$^-$ CD45.1$^+$) mice. For cell extrinsic model, $5x10^4$ of WT-OT1 (expressing CD45.2$^+$ CD45.1$^+$) splenocytes were isolated and injected into *Xiap*−/− (expressing CD45.2$^-$ CD45.1$^+$) and WT (expressing CD45.2$^+$ CD45.1$^-$) mice.

## Inhibitors and reagents

Various inhibitors were diluted from stocks in dimethyl sulfoxide (DMSO). Control cells were maintained in media containing comparable amounts of DMSO. Caspase-1 inhibitor Z-YVAD-FMK was obtained from Calbiochem (San Diego, CA). NLRP3 inhibitor (MCC950) was obtained from Cayman Chem. Co (#17510, Ann Arbor USA). LPS was obtained from Sigma Chem. Co. (St. Louis, MO). TNFα was obtained from R&D Systems (410-MT). The NOD-inhibitor MDK19922 was obtained from Selleckchem (#S0004). RipK1 inhibitor, Necrostatin-1 (Nec-1 9037) was obtained from Sigma-Aldrich).

## Western blotting

Cells were washed with cold PBS twice and lysed in 1% SDS lysis buffer with 1% β-mercaptoethanol and transferred to 1.5mL microcentrifuge tubes with cap locks. Samples were boiled

immediately at 96˚C and frozen at -20˚C for future use. Cell lysates were run on 6 – 15% poly-acrylamide gels, based on the size of the proteins of interest. Gels were typically loaded with 15 μl of lysate and run at 130 V for 85 min. The transfer was run at 75-100 V for 60-90 min. Once transferred, 5% BSA was used for 1 h to block the membrane. Primary antibodies diluted in blocking buffer (5% BSA), were added to the membrane and incubated overnight on a rocking shaker at 4˚ C.

The following antibodies were used: Mouse anti-β-actin (8H10D10) (Cell signaling, 3700S), anti-mouse MK2 (Cell signaling, 3042), anti-mouse phospho (Thr334) MK2 (Cell signaling, 3007), rabbit anti-mouse P65 (Cell signaling, 8242), rabbit anti-mouse phospho (S536) P65 (Cell signaling, 3033), rabbit anti JNK (Cell signaling, 9252), anti-mouse RipK2 (Cell signaling, 4142S), anti-mouse NOD2 (Novus Biologicals, NB100-524SS), rabbit anti-mouse phospho (T183/Y185) JNK (Cell signaling, 9251), rabbit anti-mouse phospho (Ser176/180) IKK alpha/beta (Cell signaling, 2697), rabbit anti-mouse cleaved caspase-8 (Asp387) (Cell signaling, 8592), rabbit anti-mouse cleaved caspase-9 (Asp353) (Cell signaling, 9509), rabbit anti-mouse caspase-9 (Cell signaling, 9504), rabbit anti-mouse cleaved caspase-1 (Asp296) (Cell signaling, 89332), rabbit anti-mouse cleaved caspase-3 (Asp175) (Cell signaling, 9664), rabbit anti-mouse BAD (Cell signaling, 9292), rabbit anti-mouse phospho-BAD (S26) (Abiocode, R0221-7), rabbit anti-mouse BIM (C34C5) (Cell signaling, 2933), rabbit anti-mouse IKKα (Cell signaling, 2682), rabbit anti-mouse Gasdermin D (E9S1X) (39754, Cell signaling, CA), rabbit anti-mouse phospho STAT1 (Ser727) (9177, Cell signaling, CA), rabbit anti-mouse STAT1 (9172, Cell signaling, CA); rabbit anti-mouse phospho STAT3 (Tyr705) (9131, Cell signaling, CA), and rabbit anti-mouse STAT3 (4904, Cell signaling, CA).

## Quantitative PCR

Cells were washed with cold sterile PBS (pH 7.4). RNA was prepared using RNeasy Plus Mini Kit (Qiagen, 74134) according to manufacturer's instructions. RNA concentration was determined using a Nanodrop spectrophotometer (Thermo Fisher). Complementary DNA was prepared using cDNA Synthesis Kit (Bio-Rad, 1708891) according to manufacturer's instructions. Q-RT-PCR was done using SYBR Green PCR Master Mix (Life Technologies) and the reactions were run on CFX96 Real-Time PCR System (Bio-Rad). Primer sequences are as follows:

ß-actin: (FW: GATCAAGATCATTGCTCCTCCTG; Rev: AGGGTGTAAAACGCAGCTCA)

IL-2: (FW: GCGGCATGTTCTGGATTTGACTC; Rev: CCACCACAGTTGCTGACTCATC).

## Supporting information

**S1 Fig. Comparison of immune compartments between naïve and infected WT and *Xiap* −/− mice.** Mice were infected with ST-OVA ($10^3$ CFU, i.v.). Spleens were harvested from naïve (**A**) and day 5-infected (**B**) mice, and spleen cells stained with various antibodies to identify the proportions of various immune cell subsets. Plots presented are representative of 4-5 mice.
(TIFF)

**S2 Fig. Poor antigen-presentation by *Xiap*−/− APCs.** (**A**) Expression of H2k[b] on BMDMs and BMDCs generated from WT and *Xiap*−/− mice was evaluated by flow cytometry following staining with anti-H2K[b] antibody. (**B, C**) Proliferation of CFSE labeled WT OT1 cells was evaluated at 72h post culture with ST-OVA infected BMMs. (**D**) IFN-γ production by WT OT1 cells was evaluated at 72h post culture with ST-OVA infected BMMs. (**E-G**) Bone marrow

derived DCs (WT and *Xiap*−/−) were infected with LM-OVA. Infected DCs were incubated with CFSE labeled purified WT OT1 CD8 T cells. At various time intervals, cells were harvested, stained with Zombie Yellow, and an anti-CD8 antibody and evaluated for proliferation and viability via flow cytometry (**E, F**). Cells were imaged at 72h post culture (**G**). Data is representative of 3 (experiments. Each data point represents a separate mouse. Statistical analysis was performed by 2-way ANOVA. (**P<0.01, ***P<001).
(TIFF)

**S3 Fig. Impact of XIAP on cell death and cytokine production.** (**A, G**) WT and *Xiap*−/− DCs were infected with LM-OVA grown under stationary phase (10 MOI). (**B-F**) WT and *Xiap*−/− BMMs were infected with ST (10 MOI) grown in log phase (**C**) or stationary phase (**B, D-F**). Cytokine production was evaluated at 24h in cell supernatants by ELISA. IFN-I levels were assessed by luciferase bioassay. Cell death was measured at 3h (**C**) or 24h (**D-G**) post infection by neutral red assay. Some cells were also treated with the NLRP3 inhibitor (MCC950, 10μM), RipK1 inhibitor (Nec-1, 10μM), or NOD2 inhibitor (MDK19922, 10μM). Data is representative of 3 (**A-D**, **G**) or 4 (**E, F**) experiments. Statistical analysis was performed by unpaired students t-test (*p<0.05, **p<0.01, ***p<0.001, ****P<0.0001).
(TIF)

**S4 Fig. Impact of XIAP on cell signaling.** (**A, B, E**) WT and *Xiap*−/− BMMs were infected with ST grown under stationary phase (10 MOI). Cell extracts were collected at various time intervals and the expression of various signaling proteins evaluated by western blot analysis. (**C, D**) WT and *Xiap*−/− BMMs (**C**) or DCs (**D**) were infected with various MOIs of ST grown under stationary phase. (**F**) WT and *Xiap*−/− DCs were infected with ST-OVA (10 MOI). Immediately after infection DCs were incubated with purified CFSE labeled WT OT1 cells with/without Caspase-1 inhibitor (Z-YVAD-FMK, 10 μM). Cells were harvested at 96h post co-culture, stained with anti-CD8 antibody and Zombie Yellow followed by flow cytometric evaluation of proliferation and viability. Data is representative of 3 (**A-D**) or 2 (**E, F**) experiments.
(TIFF)

**S5 Fig. Poor expansion of CD8 T cell response in *Xiap*−/− mice is not dependent on the expression of XIAP in CD8 T cells.** (**A**) Schematic representation of the adoptive transfer protocol. Splenocytes from both WT OT1 (CD45.1⁻CD45.2⁺) and *Xiap*−/− OT1 (CD45.1⁺CD45.2⁺) were mixed 1:1 and injected ($10^4$ cells each, i.v.) into *Xiap*−/− mice (CD45.1⁺ CD45.2⁻). After two days, the recipient mice were infected with ST-OVA ($10^3$ CFU, i.v.). (**B**) At day 7 post-infection, the spleens from the recipient mice were harvested and the proportions of donor OT1 cells were evaluated by flow cytometry following staining with antibodies against CD8, CD45.1 and CD45.2. (**C**) Cells were also stained with antibodies to identify various subsets of activated OT-1 CD8 T cells. Percent distribution of the CD8 T cell subsets among the transferred OT1 cells is shown. Data presented is representative of 3 (A, B) or 2 (C) experiments. Statistical analysis was performed by paired student *t*-test (*P<0.05, **P<0.01).
(TIFF)

**S6 Fig. *Xiap*−/− CD8 T cells undergo increased proliferation and secreted more IFN-γ upon stimulation by WT DCs infected with LM-OVA.** CD8 T cells were purified from WT OT1 and *Xiap*−/− OT-1 spleens and labelled with CFSE and incubated with LM-OVA infected DCs. Cells were stained with Zombie Yellow and antibodies against Ki67 and CD8 and evaluated for proliferation and viability via flow cytometry. (**A**) Representative contour plots at 72h is shown. (**B**) Percent Ki67⁺ cells among viable cells (Zombie⁻) is shown. (**C**) Representative

contour plot of zombie versus CFSE is shown. (**D**) Percent dead cells (zombie+) that have cycled ($CFSE^{low}$) is shown. (**E**) Expression of active Caspase-3 is shown in proliferating OT-1 cells. (**F**) Supernatant was collected at 24hr post infection and analyzed for IFN-γ secretion by ELISA. Data is representative of 3 (**A-D, F**) and 2 (**E**) experiments. Statistical analysis was performed by 2-way ANOVA.
(TIFF)

**S1 Data. Individual data of replicates of all the figures and supplementary files associated with this manuscript.**
(XLSX)

# Author Contributions

**Conceptualization:** Parva Thakker, Subash Sad.

**Data curation:** Parva Thakker, Ardeshir Ariana, Stephanie Hajjar, David Cai, Dikchha Rijal.

**Funding acquisition:** Subash Sad.

**Investigation:** Parva Thakker, Ardeshir Ariana, Stephanie Hajjar, David Cai, Dikchha Rijal.

**Project administration:** Subash Sad.

**Resources:** Subash Sad.

**Supervision:** Subash Sad.

**Writing – original draft:** Parva Thakker.

**Writing – review & editing:** Subash Sad.

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
