## [Decision Letter · Decision Letter 0]

26 Feb 2023

Dear Dr. Sad,

Thank you very much for submitting your manuscript "XIAP promotes the expansion and limits the contraction of CD8 T cell response through cell extrinsic and intrinsic mechanisms respectively" for consideration at PLOS Pathogens. As with all papers reviewed by the journal, your manuscript was reviewed by members of the editorial board and by several independent reviewers. In light of the reviews (below this email), we would like to invite the resubmission of a significantly-revised version that takes into account the reviewers' comments.

Thank you for submitting your work to PLOS Pathogens. The Reviewers found the study interesting but there are significant concerns that need to be address fully in a revised manuscript. Most of Reviewer 1's comments are straightforward while Reviewer 2 has concerns about data presentation and reporting technical versus biological replicates. These need to be clearly defined. Moreover, the breeding strategy to attain the mice for experiments needs to be reported - given the impact of the microbiota on immune phenotypes and especially during infection, littermate controls are required. Finally, both Reviewers found the manuscript hard to follow and unfocused. I suggest streamlining some of the findings to make the interpretations more cohesive.

We cannot make any decision about publication until we have seen the revised manuscript and your response to the reviewers' comments. Your revised manuscript is also likely to be sent to reviewers for further evaluation.

Sincerely,

Dana J. Philpott

Academic Editor

PLOS Pathogens

Nina Salama

Section Editor

PLOS Pathogens

Kasturi Haldar

Editor-in-Chief

PLOS Pathogens

orcid.org/0000-0001-5065-158X

Michael Malim

Editor-in-Chief

PLOS Pathogens

orcid.org/0000-0002-7699-2064

Thank you for submitting your work to PLOS Pathogens. The Reviewers found the study interesting but there are significant concerns that need to be address fully in a revised manuscript. Most of Reviewer 1's comments are straightforward while Reviewer 2 has concerns about data presentation and reporting technical versus biological replicates. These need to be clearly defined. Moreover, the breeding strategy to attain the mice experiments needs to be reported - given the impact of the microbiota on immune phenotypes and especially during infection, littermate controls are required. Finally, both Reviewers found the manuscript hard to follow and unfocused. I suggest streamlining some of the findings to make the interpretations more cohesive.

Reviewer's Responses to Questions

**Part I - Summary**

Reviewer #1: In their manuscript entitled „XIAP promotes the expansion and limits the contraction of CD8 T cell response through cell extrinsic and intrinsic mechanisms respectively”, Thakker and co-workers specify a novel function of XIAP in the regulation of CD8 T cell reposes towards bacterial infection.

Using elaborated T cell mouse models and genetic depletion of XIAP in combination with elegant bacterial infection models, they show that XIAP depletion leads to poor priming and increases expansion and death of CD8 cells resulting in a poor memory response. Moreover, they show that this is partly dependent on extrinsic IL-6 and enhanced cell cycle progression in XIAP ko cells.

After its identification as a negative regulator of apoptosis, XIAP was shown to also regulate NF-kB responses, other cell death events and to be an essential component of the NOD1/2 pathway. The integral complexity of the biological function of XIAP that is still not well understood. This is best exemplified by the variance of symptoms associated with disease-associated polymorphisms in XIAP.

In the present work the authors provide a detailed analysis of the function of XIAP in CD8 cell responses towards bacterial infection which is novel and provides a better understanding of role of XIAP in adaptive immunity also in general.

The manuscript is well written and contains a logical flow of very elegant studies to address the role of XIAP in CD8 cells. The reported findings are novel and well substantiated by the data. The methodology applied is very elegant and well suited to address the research question.

However, the manuscript might benefit form a somewhat clearer focus. Some parts are rather out of focus and do not provide much novel information, such as the part on antigen presenting cells. Furthermore, issues with data presentation and statistics need to be addressed.

Reviewer #2: Loss of XIAP function is the genetic cause of immunodeficiency in humans. Multiple murine studies have previously shown that myeloid cells undergo cell death in the absence of XIAP and this could lead to the hyper-inflammatory phenotypes observed in these patients. The role of XIAP in T-cells however has not been as heavily explored. Thus, in this paper, the authors examine the role of XIAP in CD8 T-cell expansion, differentiation and the development of memory. This was done specifically in the context of models for intra-cellular bacterial pathogenic infections by Salmonella enterica (ST) and Listeria monocytogenes (LM).

By performing experiments with the ST-OVA, LM-OVA and ST bacterium in both in vivo and in vitro models the authors can show that a general feature of XIAP-deficiency is that CD8 T-cells hyper-proliferate early on in response to infections, and suffer from reduced survival in the long-term. This was not always correlated with bacterial load and depended on the infection model used. As the differences were observed in the early phases of infection the authors conclude that XIAP might play a role in the priming phase.

Based on co-culture experiments with antigen presenting cells (DCs and macrophages), infected with ST-OVA and LM-OVA, WT CD8 OT1 cells proliferated differently. With Xiap-/- APC, the CD8 T-cells underwent fewer cell divisions but more cells underwent division. So the authors conclude, that XIAP-/- myeloid cells promote the rapid expansion of CD8 T cells, that then results in their rapid demise. The authors could pin-point a deficit in IL-6 production from the APCs as a contributing factor to the deregulated CD8 T cell responses, both in vivo and in vitro.

In addition to the role of XIAP-deficient APC, the authors could also demonstrate T-cell intrinsic defects of XIAP-deficient CD8 T-cells. These cells showed reduced long-term survival and reduced capacity to generate memory cells. This could be due to defects in IL-2 induction, reduced NF-KB activation and slight increases in multiple caspases. Finally, the authors looked at the long-term impact of this deficit in survival in vitro. Specifically, they look at the capacity of Xiap-/- OT1 cells to generate memory. Looking at consecutive infections of WT mice with ST-OVA and then LM-OVA, the number of Xiap-/- OT1 cells was significantly reduced compared to the WT OT1 cells. In re-infecting mice with ST-OVA the authors could also show that the Xiap-/- mice were unable to control the re-infection as well as mice with WT OT1 CD8 T-cells.

In conclusion, the authors suggest that XIAP functions to control CD8 T-cell responses via both extrinsic and intrinsic mechanisms. Extrinsically via the altered cytokine profile of infected antigen presenting Xiap-/- cells, and the intrinsic increase in early proliferation and reduced long-term survival of the CD8 T cells. This results in a deficiency in T-cell memory formation, which could also contribute to the immunodeficiency.

These aspects on CD8 biology are novel in the field of XIAP-deficiency and adds to our understanding of the disease mechanism. Much of the data regarding the APCs are not novel and could be pruned down to what is necessary for the T-cell story which is supposed to be the main subject of this manuscript.

However, I find that this paper suffers from poor structure. The result section would benefit from more detailed justifications/explanations behind the experimental setups used. I would also suggest perhaps reducing some of the redundant data for the ST-OVA and LM-OVA models that are just repetitive and do not really add novelty, but do make the text more difficult to follow.

**Part II – Major Issues: Key Experiments Required for Acceptance**

Reviewer #1: I would suggest focussing the manuscript on the characterization of the role of XIAP in CD8 cells. The part on antigen-presenting cells does not provide essential information and complicates the story. Also, the abstract would benefit by providing a clearer focus. Moreover, the part on macrophages is rather vague and lacks any detailed analysis of the molecular function of XIAP. The conclusion in line 186 that XIAP promotes IL-6 section in different infection models is not novel and at least analysis of the reported XIAP-dependent pathways, such as NOD1/2 would be wanted.

The manuscript overall is quite descriptive. No details of the molecular mechanisms, for example the role of XIAP in cell cycle, are provided. The authors should at least comment on this a bit more and provide their ideas on possible pathways.

There are critical issues with the data presentation and analysis.

All presented figures should include representation of each sample, as provided for same but not all figures. The bar graphs need to be adopted showing all data as points. It is unclear what the authors mean by “Data is representative of 2-3 experiments”. The authors need to indicate for each experiment how often this was independently performed and in how many technical replicates and what of this is shown in the graph. In order to be able to meaningful perform the indicted statistical analysis at least 3-4 independent experiments are needed. In the present form the applied statistical analysis is meaningless for most of the figures. This reviewer suggest that statistical analysis might be omitted if trends are obvious, however the underlying data of each experiment need to be graphed. Experiments that are obviously based in a single data point such in Fig.3 D need to be repeated. This also applies for the supplementary figures.

Reviewer #2: (No Response)

**Part III – Minor Issues: Editorial and Data Presentation Modifications**

Reviewer #1: - The labelling of the y-axis in Fig.1F is misleading. I would suggest to change to “Ag-specific cells”.

- spaces in front of the citations are missing in lines 287,301 and 377

Reviewer #2: With virulent ST that kills DCs and Macs in vivo infection of WT and Xiap-/- mice showed a more rapid demise of the Xiap-/- mice. This was associated with reduced CFUs in the Xiap-/- mice and increased IL1b levels and reduced IL6 in the serum. In vitro infection of WT and Xiap-/- BMM with ST also replicated the in vivo cytokine phenotype of higher IL1b and lower Il-6 production from infected Xiap-/- cells. In contrast, infection of BMM with log-phase ST showed no difference in cell death and IL1b production, while stationary phase ST infection again produced this difference in Il1b production. Could this difference in results be due to the fact that log-phase ST directly activates the pyroptotic pathway which has been shown to be intact and normal in Xiap-/- cells, while stationary phase ST is not such a strong canonical activator of pyroptosis?

Figure 2:

It would have been nice to also see the data at day 3/5 post-infection and the pre-infection control to really evaluate whether the reduction in the CD8 T cells is specific to the infection challenge, or is already visible before the infection.

Figure 4:

the constant switching between the various stages of ST for infections needs to be made clearer in the text as this impacts the interpretation of the data. For example, it is not clear what phase of ST is used in figures 4G-I.

Figure 5:

In section regarding figure 5, the authors used experiments with either ST or ST-OVA. After showing that the use of stationary or log-phase ST makes a difference in the resulting cytokine expression (Figure 4) this aspect should be made clear in all the other figures using ST and ST-OVA. In figure 5 it is not mentioned at all.

Is there an error in Fig. 5A? in the text it is stated that the Xiap-/- DCs die more but in the figure I see that WT DC die more with an MOI of 10.

S4A-B:

What is the underlying motivation for looking at TNFR1/2 blockade in this context? Was this experiment done with log/stationary phase ST? and why are such high MOIs now used here? In (B) it looks like a bit more Gasdermin D is cleaved in Xiap-/- DCs, but again why the use of such high MOIs in these experiments?

S4C:

Caspase-1 inhibition in WT and Xiap-/- DCs was performed and the impact of this on WT OT1 cell proliferation was examined. The authors state that there was no significant impact of this on CD8 T-cells, but to me it looks like Caspase1 inhibition strongly promoted the proliferation of the T-cells cultured together with WT DCs. Could the authors please comment on that? What impact did caspase-1 inhibition have on the cytokines produced by the DCs and could this explain the effects/lack of effects on the T-cells?

S4D: why is viability shown for only the MOI 0.1 and 1, but cytokines for up to MOI 10?

Could an explanation for some differences between the ST and LM models be that LM does not primarily target myeloid cells?

Line 376/377: XLP2 is an immunodeficiency.

Line 295/296: most studies examining RIPK1 within TNFR1 signaling have found no role for XIAP in ubiquitinylating of RIPK1.

PLOS authors have the option to publish the peer review history of their article (what does this mean?). If published, this will include your full peer review and any attached files.

Reviewer #1: No

Reviewer #2: No
---

## [Decision Letter · Decision Letter 1]

15 May 2023

Dear Dr. Sad,

Thank you very much for submitting your manuscript "XIAP promotes the expansion and limits the contraction of CD8 T cell response through cell extrinsic and intrinsic mechanisms respectively" for consideration at PLOS Pathogens. As with all papers reviewed by the journal, your manuscript was reviewed by members of the editorial board and by several independent reviewers. The reviewers appreciated the attention to an important topic. Based on the reviews, we are likely to accept this manuscript for publication, providing that you modify the manuscript according to the review recommendations.

Thank you for revising your manuscript according to the issues raised by the Reviewers - both Reviewers are satisfied with these revisions. However, my concern over the status of the mice used in this study was not addressed. Can the authors please indicate whether or not the mice used in the study (eg, Xiap KO vs WT) were littermate animals from a het X het cross?

Sincerely,

Dana J. Philpott

Academic Editor

PLOS Pathogens

Nina Salama

Section Editor

PLOS Pathogens

Kasturi Haldar

Editor-in-Chief

PLOS Pathogens

orcid.org/0000-0001-5065-158X

Michael Malim

Editor-in-Chief

PLOS Pathogens

orcid.org/0000-0002-7699-2064

Thank you for revising your manuscript according to the issues raised by the Reviewers - both Reviewers are satisfied with these revisions. However, my concern over the status of the mice used in this study was not addressed. Can the authors please indicate whether or not the mice used in the study (eg, Xiap KO vs WT) were littermate animals from at het X het cross?

Reviewer Comments (if any, and for reference):

Reviewer's Responses to Questions

**Part I - Summary**

Reviewer #1: The authors sufficiently addressed my queries. The revised manuscript is now somewhat clearer structured and in principal suitable for publication in my view.

Reviewer #2: The authors have satisfactorily addressed the questions and issues raised at the first submission.

**Part II – Major Issues: Key Experiments Required for Acceptance**

Reviewer #1: (No Response)

Reviewer #2: (No Response)

**Part III – Minor Issues: Editorial and Data Presentation Modifications**

Reviewer #1: (No Response)

Reviewer #2: (No Response)

PLOS authors have the option to publish the peer review history of their article (what does this mean?). If published, this will include your full peer review and any attached files.

Reviewer #1: No

Reviewer #2: No

Figure Files:

Data Requirements:

Reproducibility:

References:

---

## [Editor Report · Decision Letter 2]

1 Jun 2023

Dear Dr. Sad,

We are pleased to inform you that your manuscript 'XIAP promotes the expansion and limits the contraction of CD8 T cell response through cell extrinsic and intrinsic mechanisms respectively' has been provisionally accepted for publication in PLOS Pathogens.

Best regards,

Dana J. Philpott

Academic Editor

PLOS Pathogens

Nina Salama

Section Editor

PLOS Pathogens

Kasturi Haldar

Editor-in-Chief

PLOS Pathogens

orcid.org/0000-0001-5065-158X

Michael Malim

Editor-in-Chief

PLOS Pathogens

orcid.org/0000-0002-7699-2064

Congratulations on your paper!
---

## [Editor Report · Acceptance letter]

17 Jun 2023

Dear Dr. Sad,

We are delighted to inform you that your manuscript, "XIAP promotes the expansion and limits the contraction of CD8 T cell response through cell extrinsic and intrinsic mechanisms respectively," has been formally accepted for publication in PLOS Pathogens.

Best regards,

Kasturi Haldar

Editor-in-Chief

PLOS Pathogens

orcid.org/0000-0001-5065-158X

Michael Malim

Editor-in-Chief

PLOS Pathogens

orcid.org/0000-0002-7699-2064